# New Quinoxaline-Based Derivatives as PARP-1 Inhibitors: Design, Synthesis, Antiproliferative, and Computational Studies

**DOI:** 10.3390/molecules27154924

**Published:** 2022-08-02

**Authors:** Yasmin M. Syam, Manal M. Anwar, Somaia S. Abd El-Karim, Khaled M. Elokely, Sameh H. Abdelwahed

**Affiliations:** 1Department of Therapeutic Chemistry, National Research Center, Dokki, Cairo 12622, Egypt; manal.hasan52@live.com (M.M.A.); somaia_elkarim@hotmail.com (S.S.A.E.-K.); 2Institute for Computational Molecular Science, Department of Chemistry, Temple University, Philadelphia, PA 19122, USA; kelokely@temple.edu; 3Department of Chemistry, Prairie View A&M University, Prairie View, TX 77446, USA

**Keywords:** quinoxaline, PARP-1 inhibitor, antiproliferative, MDA-MB-436, WI-38, cell cycle, apoptosis, autophagy, molecular docking, ADME parameters

## Abstract

Herein, 2,3-dioxo-1,2,3,4-tetrahydroquinoxaline was used as a bio-isosteric scaffold to the phthalazinone motif of the standard drug Olaparib to design and synthesize new derivatives of potential PARP-1 inhibitory activity using the 6-sulfonohydrazide analog **3** as the key intermediate. Although the new compounds represented the PARP-1 suppression impact of IC_50_ values in the nanomolar range, compounds **8a**, **5** were the most promising suppressors, producing IC_50_ values of 2.31 and 3.05 nM compared to Olaparib with IC_50_ of 4.40 nM. Compounds **4**, **10b**, and **11b** showed a mild decrease in the potency of the IC_50_ range of 6.35–8.73 nM. Furthermore, compounds **4**, **5**, **8a**, **10b**, and **11b** were evaluated as in vitro antiproliferative agents against the mutant BRCA1 (MDA-MB-436, breast cancer) compared to Olaparib as a positive control. Compound **5** exhibited the most significant potency of IC_50_; 2.57 µM, whereas the IC_50_ value of Olaparib was 8.90 µM. In addition, the examined derivatives displayed a promising safety profile against the normal WI-38 cell line. Cell cycle, apoptosis, and autophagy analyses were carried out in the MDA-MB-436 cell line for compound **5,** which exhibited cell growth arrest at the G2/M phase, in addition to induction of programmed apoptosis and an increase in the autophagic process. Molecular docking of the compounds **4**, **5**, **8a**, **10b**, and **11b** into the active site of PARP-1 was carried out to determine their modes of interaction. In addition, an in silico ADMET study was performed. The results evidenced that compound **5** could serve as a new framework for discovering new potent anticancer agents targeting the PARP-1 enzyme.

## 1. Introduction

Poly (ADP-ribose) polymerases (PARPs) constitute a group of at least 17 enzymes that are correlated to the DNA damage repair process. PARP-1is the most abundant member of this group and has emerged as one of the most auspicious molecular targets for cancer management in the past decade [1,2,3,4]. PARP-1 acts as a “molecular nick sensor” to DNA single-strand (ssDNA) breaks and catalyzes the transference of ADP-ribose units (utilizing nicotinamide adenine dinucleotide (NAD^+^) as a substrate) to acceptor proteins, facilitating the recruitment of the damaged DNA and promoting cell survival. It is an important stage in the base excision repair (BER) of single-strand DNA breaks [4], which is linked to the resistance that typically develops following traditional cancer treatments [5,6,7]. PARP-1 suppression enhances the damage of injured DNA resulting in synthetic lethality in DNA-repairing-deficient cancer cells, such as BRCA1/2-deficient cells. Thus, PARP-1 suppression synergizes the impact of various antiproliferative drugs such as topoisomerase-I inhibitors and DNA alkylating drugs in addition to ionizing radiation. Moreover, some PARP suppressors are effective as single agents against cancers bearing BRCA1- or BRCA2-mutations [8,9,10,11].

The US FDA recently approved four PARP suppressors, Olaparib, Rucaparib, Niraparib, and Talazoparib, for curing BRCA-mutated, HER2-negative advanced, metastatic ovarian, or breast cancer. In addition, there are a number of PARP suppressors that are under study in various clinical phases such as Veliparib, Pamiparib, Simmiparib, and Fluzoparib [12,13] (Figure 1). Furthermore, recent studies investigated the therapeutic potential of various PARP-1 suppressors for other refractory diseases such as Alzheimer’s disease (AD) [14,15]. Accordingly, the development of effective PARP-1 inhibitors plays an important role in medicinal chemistry communities.

It has been reported that the catalytic pocket of PARP1 is divided into three sub-pockets that are occupied by the substrate NAD^+^. The first sub-pocket is the nicotinamide-ribose binding site (NI site), the second is the phosphate-binding site (PH site), and the third is the adenine-ribose binding site (AD site) [16] (Figure 2).

It has been reported that most of the PARP-1 suppressors bind with the NI site via H-binding and π-π stacking interactions, and some of them produce further interactions in the adenine-ribose binding (AD) site, which is large enough to fit a variety of molecules, leading to enhancing their effectiveness and pharmacokinetic characteristics [17,18]. Many studies have determined that the design of PARP-1 inhibitors is based on the nicotinamide section of NAD^+^ to imitate the ligand-protein binding of NAD^+^ with PARP-1 [19]. Accordingly, PARP-1 suppressors shar common pharmacophoric features, which are an aromatic ring and a carboxamide core. The critical bindings between them are the H-bonding networks initiated between the carboxamide moiety and Gly863 (NH to Gly C=O and C=O to Gly NH) and Ser904 (C=O to Ser OH). Additionally, the phenyl ring of Tyr907 induces the π-π stacking interaction with the aryl ring. Additionally, an auxiliary appendage with a linking side chain is commonly conjugated with the polycyclic core as a solvent accessory region in the AD site [20,21,22,23] (Figure 3).

Plenty of research has shown that the improvement of PARP inhibitors’ binding affinity by restriction of the carboxamide’s free rotation greatly enhances the PARP1 inhibitory activity. The carboxamide moiety can be locked into the required confirmation by either inserting the aromatic ring heteroatoms or functionalities that can form an intramolecular hydrogen bond with the amide NH or conjugating the amide group in a bicyclic system [4,6,24].

Quinoxaline is a privileged scaffold and one of the main blocks of different anticancer agents as it has been proven to be selective adenosine triphosphate (ATP) competitive as well as a bioisostere to benzimidazole, quinazolinones, isoquinolinones, phenanthridone, or phthalazinones, which are the basic scaffolds of the plurality of PARP-1 inhibitors [25,26,27]. In addition, sulfonyl and sulfonamide moieties conjugated to different heterocyclic ring systems have been reported as one of the most privileged scaffolds to inhibit the growth of various human cancer cell lines via different modes of action [28,29]. Cancer treatment is still a challenge due to the development of cancer cell resistance, toxicity, and the lack of selectivity of most commercialized anticancer medications. As a result, and in view of the continuation of our efforts in discovering new heterocyclic compounds of potential anticancer activity targeting the PARP-1 enzyme [30,31,32], the strategy of this study was focused on the design and synthesis of the new compounds based on the quinoxaline core to occupy the NI site of PARP-1 hybridized at its position-6 with different heterocycles, such as pyrrole, pyrazole, thiazole, imidazolidinone, and pyrimidine via sulfonyl, sulfonamide, and sulfonohydrazide linkers aiming to engage with the PARP-1 enzyme through different binding modes of action. The quinoxaline nucleus bears two carboxamide moieties that engage with the enzyme through additional hydrogen bonding (Figure 3). All new compounds were examined as PARP-1 inhibitors. Since PARP-1 inhibitors result in synthetic lethal effects, specifically in BRCA-mutated cells, MDA-MB-436 (BRCA-1-mutated breast cancer cell line) was selected to conduct a cell proliferation assay for the analogs that exhibited the most active inhibition effect against the PARP-1 enzyme. Thereafter, the safety margin of the most potent members was evaluated against WISH normal cells. A molecular docking study was also employed for the promising PARP-1 inhibiting candidates to rationalize and emphasize their mechanisms of binding with the active pocket of the target enzyme. Furthermore, in silico ADMET prediction was performed for the new compounds to explore their drug-likeness characteristics.

## 2. Results

### 2.1. Chemistry

This study was directed toward the design and construction of new quinoxaline compounds using various synthetic pathways illustrated in Figure 1 and Figure 2. The synthesis was initiated by reacting the starting material *o*-phenylenediamine with oxalic acid in the presence of HCl to provide quinoxaline-2,3(1*H*,4*H*)-dione (**1**), which was treated with chlorosulfonic acid to provide the corresponding key intermediate 6-sulfonyl chloride derivative **2** according to the reported methodology [33,34]. The latter derivative served as a facile intermediate for the nucleophilic substitution reaction with hydrazine hydrate to afford the 6- sulfonohydrazide derivative **3****,** which was utilized as a precursor for the ring closure reaction with different active methylene reagents, namely, ethyl-acetoacetate, acetylacetone, and diethyl malonate, to accomplish the corresponding pyrazole derivatives **4**–**6**, respectively.

IR spectra of compounds **3**–**6** demonstrated stretching bands at approximately 3441–3315 cm^−1^ corresponding to the NH_2_ and NH groups, other bands ranging from 1750 to 1678 cm^−1^ due to C=O groups, and at 1392–1138 corresponding to SO_2_ groups.

^1^H NMR spectra of compounds **3**–**6** represented D_2_O exchangeable signals of NH_2_ and NH functionalities in the range of δ 11.78–11.83 ppm, alongside multiplet signals at the region of δ 7.01–8.03 ppm related to the aromatic protons. The sulfonohydrazide compound **3** showed an additional D_2_O exchangeable singlet at δ 7.85 ppm assignable to the NH_2_ group, while the target pyrazole derivatives **4**, **5**, and **6** represented new singlets at δ 2.12–2.74 ppm related to CH_3_, 2CH_3_, and CH_2_ functionalities, respectively, at δ 6.78–6.31 ppm due to the pyrazole-H_4_ of compounds **4** and **5**, and at 12.35 ppm exchangeable with D_2_O related to the OH group of compound **4**. Furthermore, ^13^C NMR spectra of compounds **4**, **5**, and **6** exhibited singlet signals at δ 12.53, 11.21, and 55.92 ppm assignable to CH_3_, 2CH_3_, and CH_2_ groups, respectively, at δ 102.07–155.73 related to the aromatic carbons, and at δ 154.82–167.89 ppm due to C=O groups.

The further condensation reaction of 6- sulfonohydrazide derivative **3** with different acid anhydrides, namely succinic, maleic, and/or phthalic anhydride in glacial acetic acid, resulted in the achievement of the corresponding analogs **7a**–**c**, respectively. ^1^H NMR spectra of the latter derivatives **7a**–**c** revealed multiplet signals in the range of δ 7.32–7.93 ppm, contributing to the aromatic protons, as well as three D_2_O exchangeable signals at the region δ 9.61–12.01 ppm due to 3NH groups. Compound **7a** represented a singlet signal at δ 2.73 ppm corresponding to the dioxopyrrolidine-2CH_2_ function, **7b** exhibited a doublet signal at δ 7.20 ppm attributed to its vinylic protons, while **7c** revealed an increase in the integration values in the aromatic region due to the phthalic protons. Moreover, ^13^C NMR spectra of compounds **7a**–**c** exhibited singlet signals in the range of δ 115.13–138.42 ppm assigned to the aromatic carbons and in the range of δ 154.34–170.81 ppm representing C=O groups. A singlet signal appeared at δ 30.07 ppm due to the two methylene carbons of the pyrrolidine-2CH_2_ of compound **7a** (Figure 1).

Furthermore, the reaction of 6-sulfonohydrazide derivative **3** with 4-methoxybenzene isothiocyanate and/or benzoyl isothiocyanate in refluxing DMF in the presence of a few drops of triethylamine resulted in the formation of the thiosemicarbazide derivatives **8a**,**b**, respectively. ^1^H NMR spectra of the latter derivatives **8a**,**b** represented singlet signals at δ 10.71–11.97 ppm exchangeable with D_2_O affordable to NH groups, multiplet signals at δ 6.90–8.14 ppm related to the aromatic protons, and a singlet signal at δ 3.81 ppm assignable to the methoxy protons (-OCH_3_) in the case of compound **8a**. Furthermore, their ^13^C NMR spectra revealed the parent carbons of both derivatives in companion with a singlet signal at δ 55.44 ppm related to OCH_3_ of **8a** and at δ 163 ppm due to ph-C=O of the benzoyl compound **8b**.

Thiosemicarbazide congeners are reported to be valuable intermediates in organic chemistry since they act as building blocks for the preparation of various heterocyclic compounds possessing biological importance [35,36]. Accordingly, the treatment of compounds **8a**,**b** with diethyl malonic acid in refluxing ethanol furnished the corresponding 2-thioxo-3,4-dihydropyrimidine derivatives **9a**,**b**, respectively. IR spectra of compounds **9a**,**b** displayed different absorption bands at 3441–3160, 1710–1645, 1415, 1338, and 1196 cm^−1^ related to 3NH, 3C=O, C=S, and SO_2_ groups. ^1^H NMR spectra of compounds **9a**,**b** represented an additional signal at δ 10.51 ppm exchangeable with D_2_O referring to the OH group, while the pyrimidine-H_5_ appeared as a singlet signal in the aromatic region at δ 7.12–7.35 ppm alongside the parent protons, which appeared in their expected regions. Furthermore, ^13^C NMR spectra of compounds **9a**,**b** represented singlet signals at the region of δ 82.50–155.47 ppm related to the aromatic carbons, at the region of δ 155.40–155.50 due to C=O groups, and at δ 170.1, 189.68 ppm due to C=S groups. The methoxy carbon of compound **9a** appeared as a singlet signal at δ 55.80 ppm.

Moreover, the nucleophilic reaction of various α- halo ketones, namely chloroacetone and 3-chloroacetylacetone with compounds **8a**,**b,** was carried out in the presence of sodium acetate to give the corresponding thiazolines, **10a**,**b**, and **11a**,**b**, respectively. ^1^H NMR spectra of compounds **10a**,**b** exhibited a singlet signal in the region of δ 1.85–2.31 ppm contributing to the thiazoline-CH_3_ protons, while the thiazoline-H_5_ appeared as a singlet signal at δ 5.45 ppm, in addition to the precursor protons, which was presented in the correct regions. Furthermore, ^1^H NMR spectra of **11a**,**b** exhibited two singlet signals at δ 2.33 and 2.51 ppm assigned to thiazoline-CH_3_ and COCH_3_, respectively, alongside the signals of the parent protons. Similarly,^13^C NMR spectra of **10a**,**b**, and **11a**,**b** represented CH_3_ carbons as singlet signals in the region of δ 14.56–19.77 ppm, in addition to the parent carbons, which were presented in their correct regions. The acetyl carbon of compounds **11a**,**b** was represented as a singlet signal at δ 25.13 and 25.62 ppm, respectively.

Moreover, the treatment of **8a**,**b** with ethyl bromoacetate in absolute ethanol containing a catalytic amount of sodium acetate accomplished the corresponding thiazolidine derivatives **12a**,**b**, respectively. ^1^H NMR spectra of the target derivatives **12a**,**b** showed an up-field signal in the region of δ 4.67 ppm corresponding to the thiazolidine-CH_2_ methylene protons alongside the parent protons, which appeared at their expected regions. ^13^C NMR of the latter derivatives showed a singlet signal at δ 25.13 assignable to the thiazolidine-CH_2_, and singlet signals in the regions of δ 114.66–155.68 ppm and 163.27–176.33 due to the aromatic and C=O carbons, respectively. The methoxy carbon of **12a** appeared as a singlet signal at δ 63.09 ppm (Figure 2). Mass spectra of the newly prepared compounds exhibited correct molecular ion peaks, which were in accordance with their molecular formulae.

### 2.2. Biological Evaluation

#### 2.2.1. Assessment of PARP-1 Inhibitory Activity of the Target Quinoxaline Compounds

All the new target quinoxaline compounds **3**–**12a**,**b** were evaluated as PARP-1 inhibitors to gain clear insight into the structure–activity relationship using the colorimetric 96-well PARP-1 assay kit [6,35]. The IC_50_ values of all the tested compounds against PARP-1 were expressed in nM concentrations utilizing Olaparib as a reference drug and are summarized in Table 1. Despite all the obtained IC_50_ values being in the nanomolar range, they showed a wide variation in PARP-1 inhibitory activity (IC_50_ range; 2.31–57.35 nM). Therefore, it can be supposed that the terminal position of the molecule can tolerate a wide variety of substituents and this point can be explained if these side chains are approaching the solvent surface and do not bind significantly with the enzyme. The key starting intermediate 6-sulfonohydrazide derivative **3** displayed 3-fold less inhibitory activity against PARP-1 than that of the reference drug Olaparib of IC_50_s; 12.86, 4.40 nM, respectively. We detected the direct conjugation of the parent 6-sulfonoquinoxaline core with a 3,5-dimethylpyrazole ring as compound **5** exhibited PARP-1 inhibitory activity higher than that of the control drug by 1.5 folds (IC_50s_ = 3.05 nM). Conversely, the replacement of either one or both methyl groups of the pyrazole ring of compound **5** with OH or 2C=O groups as compounds **4** and **6**, respectively, decreased the suppression impact by nearly 2- and 3-fold of IC_50s_ = 8.73, 13.27 nM, respectively. This result indicated that the hydrophobic residues are favorable for PARP-1 inhibition activity.

Moreover, hybridization of the 6-sulfonoquinoxaline scaffold with the *p*-methoxyphenyl ring via the thiosemicarbazide linker as compound **8a** represented a promising impact on the PARP-1 suppression effect, nearly 2-fold higher than that of Olaparib of IC_50_ = 2.31 nM. On the other hand, the inhibitory activity was 2.5 times less than the reference drug upon conjugation of the thiosemicarbazide side chain with the CO-unsubstituted phenyl ring as compound **8b** of IC_50_ = 11.06 nM. The *p*-substitution of the phenyl ring with the electron-donating group OCH_3_ signified the inhibitory potency as depicted by compound **8a**.

On the other hand, the hybridization of the parent 6-sulfonoquinoxaline with 3-benzoyl-4-methylthiazoline and 5-acetyl- 3-benzoyl-4-methylthiazoline moieties via the hydrazide linker as congeners **10b**, **11b** produced a slight reduction in the inhibitory activity compared to the reference drug of IC_50s_ = 6.35, 8.25 nM, respectively. A detectable decrease in the activity was further observed by the 4-methoxyphenyl analogs **10a** and **11a** of IC_50s_ = 21.63 and 19.45, respectively. It could be noted that the decrease in the thiosemicarbazide length is not favorable for the potency of the desired activity. With respect to the series of pyrrole and isoindoline derivatives **7a**–**c**, the 2-thioxopyrimidine derivatives **9a**,**b** and the 4-oxothiazolidine derivatives **12a**,**b** exhibited the lowest activity of IC_50_ values ranging from 35.82–57.14 nM. The SAR study of the most potent active congeners is depicted in Figure 4.

#### 2.2.2. Antiproliferative Activity

In order to find out the relationship between the anticancer potency and the PARP-1 suppression effect, the most effective compounds as PARP-1 inhibitors (**4**, **5**, **8a**, **10b**, **11b**) were further evaluated for their in vitro cytotoxicity against the mutant BRCA1 (MDA-MB-436, breast cancer) using an MTT assay [37]. Olaparib was used as a positive control. The IC_50_ values of all examined compounds are tabulated in Table 1. The resultant data exhibited that the dimethyl pyrazole compound **5** exhibited the best antiproliferative activity against the examined cancer cell line, being approximately 4 times more potent than the reference drug with IC_50_ values of 2.57, 8.90 µM, respectively. Furthermore, the tested compounds **8a**, **10b**, and **11b** exhibited an approximately equal activity to that of Olaparib with IC_50_ values of 10.70, 9.62, and 11.50 µM, respectively. On the contrary, compound **4** displayed the weakest antitumor activity with an IC_50_ value of 30.30µM.

This result represented an outstanding correlation between PARP-1 suppression activity and the anticancer activity of the tested compounds.

It has been reported that the frequency and severity of the side effects on normal healthy cells at therapeutic levels are deemed to be critical factors that distinguish different anticancer drugs from each other. Accordingly, the cytotoxic activity of the potent members **5**, **8a**, **10b**, and **11b** was evaluated against the normal WI-38 cell line via an MTT assay to determine their safety profiles. It is worth mentioning that the IC_50_ values of all the representative compounds against the normal cells range from 70.46–81.67 µM, which are 7–8-fold higher than their IC_50s_ values against the cancer cell line, confirming their promising safety profile (Table 1).

#### 2.2.3. Cell Cycle Analysis in MDA-MB-436

Based on its well-balanced biological activity, i.e., promising PARP-1 inhibition and high antiproliferative activity, compound **5** was chosen as a representative example for further examining cellular mechanisms with respect to its impact on cell cycle progression and induction of apoptosis in MDA-MB-436 cells by using the flow cytometric technique [38,39]. In the present work, MDA-MB-436 cells were treated with compound **5** at its IC_50_ concentration of 2.57 µM and incubated for 48 h. The obtained results were compared to the results obtained by MDA-MB-436 cells incubated with dimethylsulfoxide for 48 h as a negative control. The obtained data are summarized in Figure 5 and Figure 6.

Figure 5 represents a decrease in the cell distribution of MDA-MB-436 in the G1 phase from 51.96% (control) to 40.89% (**5**-treated cells) with a concomitant rise in the percentage of cells in the G2 stage from 25.12% (control) to 30.39% (**5**-treated cells), which proves that compound **5** arrested the MDA-MB-436 cell cycle at the G2/M stage. In addition, the cell number in the sub-G1 stage was 1.76% (control cells) and 1.66% in (**5**-treated cells).

#### 2.2.4. Apoptosis Assay in MDA-MB-436 Cells

Annexin V-FITC and PI staining coupled with flow cytometry was utilized for further investigation of the apoptotic impact of compound **5** on MDA-MB-436. Treatment of MDA-MB-436 cells with the IC_50_ concentration of **5** induced both apoptosis and necrotic effects (Figure 7 and Figure 8). The early apoptotic cell population increased from 0.14% (control) to 0.27% (**5**-treated cells), the percentage of the late apoptotic cell population increased from 0.98% (control) to 1.02% (**5**-treated cells), and the necrotic cell population increased from 1.11% (control) to 1.72% (**5**-treated cells).

#### 2.2.5. Autophagy Assay

It has been reported that autophagy-induced programmed cell death is a hot topic in the scientific community. It was of interest to study the effect of compound **5** on the autophagy process within MDA-MB-436 cells utilizing Cyto-ID autophagy detection dye coupled with flow cytometry [40,41]. Treatment of the latter cells with **5** increased autophagic cell death by 68.65% (Figure 9).

### 2.3. Computational Studies

#### 2.3.1. Molecular Docking

To find out the interaction modes of the most promising quinoxaline congeners **4**, **5**, **8a**, **10b**, and **11b**, a standard docking protocol was used where Olaparib was utilized as the reference frame of the docking grid. The crustal structure of Olaparib in complex with the catalytic domain of PARP1 was downloaded from the protein databank (www.rcsb.org accessed 15 April 2021). The complex structure was processed with the Protein Preparation Wizard in Maestro to add missing atoms, sidechains, and residues, complete loops, add hydrogen atoms, and adjust bond orders for amino acids and ligands [42,43,44,45,46].

The five compounds were docked with high affinity, and the prime MM-GBSA free energy of binding was computed as −93 kcal/mol for Olaparib, −79.3 kcal/mol for compound **8a**, −63.9 for compound **10b**, −60.9 kcal/mol for compound **5**, −54.4 kcal/mol for compound **11b**, and −54.3 kcal/mol for compound **4**. The compounds fit well in the binding pocket and demonstrated several favorable interactions with the surrounding amino acids. Olaparib is complex with PARP1 in the crystal structure, and it showed the following interactions: Hydrogen bonds with Ser904, Gly863, Ser864, and Tyr896, water-bridged hydrogen bonds with Arg878, Ile879, and π-π contacts with Tyr896 and Tyr907. The compounds showed the following interactions: Compound **8a** interacts with hydrogen bonds with Ser904, His862, Asp766, Ser864, π-π contacts with Tyr907 and His862, and cation-π contacts with Arg848; compound **5** showed hydrogen bonds with Gly863 and Ser904, and π-π contacts with Tyr907; compound **10b** interacts through hydrogen bonds with Tyr896 and Ser894, a water-bridged hydrogen bond with Arg873, and π-π contacts with Tyr907 and Tyr889; compound **11b** showed hydrogen bonds with Ser904, Asn906, and Lys903, water-bridged hydrogen bonds with Met890 and Glu988, π-π contacts with Tyr907, and cation-π contacts with Lys903; and compound **4** interacted through hydrogen bonds with Ser904, Gly863, and Asp766, and π-π contacts with Tyr907 and Tyr889 (Figure 10 and Figure 11 and Table 2).

#### 2.3.2. Prediction ADME parameters

The SwissADME tool was used to calculate the physicochemical properties of the tested compounds **4**, **5**, **8a**, **10b**, and **11b** (Figure 12) [47,48,49,50,51,52,53]. The compounds showed low to moderate water solubility. Compound **4** showed the highest predicted solubility. Only compound **11b** violated Lipinski’s role of five having more than 10 NH and OH groups. All the compounds were computed to have low GIT absorption with the exception of compound **5,** which has a promising property. All the compounds are not expected to cross the BBB.

## 3. Materials and Methods

### 3.1. Chemistry

The instruments used for measuring the melting points, spectral data (IR, Mass, ^1^H NMR, and ^13^C NMR), and elemental analyses are provided in detail in Appendix A.

### 3.2. PARP-1 Inhibition Assay

PARP-1 enzyme inhibition activity was evaluated using a colorimetric 96-well PARP-1 assay kit (catalog no. 80580) (BPS Bioscience), according to the manufacturer’s protocol. More details are provided in Appendix A.

### 3.3. In Vitro Anticancer Screening

The in vitro cytotoxicity potency was screened against the MDA-MB-436 cancer cell line by MTT assay. The cytotoxicity was estimated as IC_50_ in µM for the tested compounds and the reference drug Olaparib. More details are provided in Appendix A.

### 3.4. Cell Cycle Analysis

The pre-calculated IC_50_ of compound **5** was applied to MDA-MB-436 breast cancer cells for 48 h. The cells were treated with trypsin, rinsed two times in PBS, fixed in ice-cold 60% ethanol at 40 °C, and washed again in PBS. More details are provided in Appendix A.

### 3.5. Apoptosis Analysis

MDA-MB-436 cells were treated with compound **5** for 48 h, then treated with trypsin and rinsed twice in PBS. Apoptosis assessment was performed via the “Annexin V-FITC/PI Apoptosis Detection Kit”, “BD Biosciences, San Diego, CA, USA”, as stated by the manufacturer. More details are provided in Appendix A.

### 3.6. Autophagy Analysis

To further confirm the cell death mechanism induced by the drugs, autophagic cell death was quantitatively analyzed using a Cyto-ID Autophagy Detection Kit (Abcam Inc., Cambridge Science Park, Cambridge, UK). More details are provided in Appendix A.

### 3.7. Docking Methodology

The crustal structure of Olaparib in complex with the catalytic domain of PARP1 [42] was downloaded from the protein databank (www.rcsb.org accessed on 15 April 2021). The complex structure was processed with the Protein Preparation Wizard [43,44] in Maestro [45] to add missing atoms, sidechains, and residues, complete loops, add hydrogen atoms, and adjust bond orders for amino acids and ligands. More details are provided in Appendix A.

### 3.8. Chemical Synthesis

#### 3.8.1. Preparation of Quinoxaline-2,3(1H,4H)-dione (**1**)

A mixture of *o*-phenylenediamine (5.0 g, 46.3 mmol) in 100 mL 4N HCl and oxalic acid (4.17 g, 46.3 mmol) was heated under reflux for 4 h. Then the reaction mixture was cooled to room temperature and the resulting solid was filtered and washed with ethanol to give the required compound **1** as a white solid according to the reported method [33]. Yield 91.5%, m.p. > 300 °C

#### 3.8.2. Preparation of 2,3-Dioxo-1,2,3,4-tetrahydroquinoxaline-6-sulfonyl Chloride (**2**)

The starting compound 1,4-quinoxaline-2,3-dione (**1**) (1.90 g, 10 mmol) was added portion-wise to chlorosulfonic acid (2 mL, 3 mmol) at 65–90 °C over 3 h. The reaction mixture was cooled to room temperature and poured slowly onto the ice/water mixture. The formed precipitate was collected by filtration and washed with water and dried. The obtained product was crystallized from benzene/petroleum ether (40–60) to give the desired 6-sulfonyl chloride product as a yellowish-white solid according to the reported method [34]. Yield 75%, m.p. 280 (decomposed).

#### 3.8.3. Preparation of 2,3-Dioxo-1,2,3,4-tetrahydroquinoxaline-6-sulfonohydrazide (**3**)

A solution mixture of compound **2** (2.60 g, 10 mmol) and hydrazine hydrate 98% (40 mmol, 2 mL) in ethanol (30 mL) was stirred at room temperature for 7 h. The obtained ppt was filtered and crystallized from ethyl alcohol to give the target 6-sulfonohydrazide as a white powder.

Yield (65%); mp. 285–287 °C; IR (KBr, cm^−1^): 3420 (NH_2_, forked), 3344–3320 (3NH), 3167 (CH, aromatic), 2990 (CH-alicyclic), 1750 (2C=O), 1332, 1138 (SO_2_); ^1^H NMR (DMSO-*d_6_*, δ ppm): 7.01, 7.41 (2d, 2H, aromatic-H, *J* = 10.03 Hz), 7.85 (s, 2H, NH_2_^,^ D_2_O exchangeable), 8.03 (s, 1H, aromatic-H), 8.63, 9.74, 10.01 (3s, 3H, 3NH, exchangeable with D_2_O); ^13^C NMR (DMSO-*d_6_*, δ ppm): 120.29, 123.47, 126.14, 129.05, 133.60, 138.63 (aromatic-C), 155.66 (2C=O)MS, *m*/*z* (%): 258 [M^+^ + 2] (35.08), 256 [M^+^] (11.06); Analysis for C_8_H_8_N_4_O_4_S (256.24), Calcd.: %C, 37.50; H, 3.15; N, 21.78; S, 12.51; Found: %C, 37.62; H, 2.93; N, 21.93; S, 12.73.

#### 3.8.4. Preparation of 6-(Substituted pyrazolyl)-sulfonyl-1,4-dihydroquinoxaline-2,3-dione Derivatives **4**–**6**

A mixture of the sulfonohydrazide compound **3** (2.56 g, 10 mmol) with different active methylene reagents, namely ethyl acetoacetate, acetylacetone, and/or diethyl malonate, was refluxed in acetic acid (15 mL) for 8 h. The formed ppt was filtered after cooling, dried, and crystallized from the proper solvent to give the corresponding compounds **4**, **5**, and **6**, respectively.

##### 6-((5-Hydroxy-3-methyl-1H-pyrazol-1-yl)sulfonyl)-1,4-dihydroquinoxaline-2,3-dione (**4**)

Yield (70%); mp. 274–276 °C, IR (KBr, cm^−1^): 3441 (OH), 3132 (NH), 3059 (CH, aromatic), 2927 (CH-alicyclic), 1687 (2C=O), 1392, 1157 (SO_2_). ^1^H NMR (DMSO-*d_6_*, δ ppm): 2.12 (s, 3H, CH_3_), 7.12 (s, 1H, pyrazole-H), 7.11 (d, 1H, aromatic-H, *J* = 8.01 Hz), 7.21–7.27 (m, 2H, aromatic-H), 11.78, 11.83 (2s, 2H, 2NH, exchangeable with D_2_O), 12.34 (s, 1H, OH, exchangeable with D_2_O); ^13^C NMR (DMSO-*d_6_*, δ ppm): 12.53 (CH_3_), 102.07, 119.08, 121.09, 123.52, 127.25, 128.37, 140.39, 142.18, 155.66, 155.73 (aromatic-C), 160.75 (2C=O); MS, *m*/*z* (%): 323 [M^+^ + 1] (25.48), 322 [M^+^] (17.07). Analysis f3or C_12_H_10_N_4_O_5_S (322.30), Calcd.: %C, 44.72; H, 3.13; N, 17.38; S, 9.95. Found: %C, 44.69; H, 3.37; N, 17.48; S, 10.26.

##### 6-((3,5-Dimethyl-1H-pyrazol-1-yl)sulfonyl)-1,4-dihydroquinoxaline-2,3-dione (**5**)

Yield (68%); mp. 268–270 °C; IR (KBr, cm^−1^): 3344–3315 (3NH), 3062 (CH, aromatic), 2958 (CH-alicyclic), 1685 (2C=O), 1381, 1180 (SO_2_); ^1^H NMR (DMSO-*d_6_*, δ ppm): 2.31 (s, 6H, 2CH_3_), 6.31 (s,1H, pyrazole-H), 7.04, 7.31 (2d, 2H, aromatic-H, *J =* 8.01 Hz), 7.45 (s, 1H, aromatic-H), 11.94 (s, 2H, 2NH, exchangeable with D_2_O); ^13^C NMR (DMSO-*d_6_*, δ ppm): 11.21 (2CH_3_), 106.67, 113.21, 114.78, 121.06, 125.17, 126.14, 143.66, 145.33 (aromatic-C), 155.66, 155.73 (2C=O); MS, *m*/*z* (%): 320 [M^+^] (33.67); Analysis for C_13_H_12_N_4_O_4_S (320.32), Calcd.: %C, 48.75; H, 3.78; N, 17.49; S, 10.01; Found: %C, 48.93; H, 3.85; N, 17.43; S, 9.86.

##### 6-((3,5-Dioxopyrazolidin-1-yl)sulfonyl)-1,4-dihydroquinoxaline-2,3-dione (**6**)

Yield (65%); mp. 270–272 °C; IR (KBr, cm^−1^): 3344–3320 (3NH), 3132 (CH, aromatic), 2999 (CH-alicyclic), 1750, 1678 (4C=O), 1332, 1138 (SO_2_); ^1^H NMR (DMSO-*d_6_*, δ ppm): 3.36 (s, 2H, CH_2_), 7.01–7.08 (m, 2H, aromatic-H), 7.41 (s, 1H, aromatic-H), 9.74 (1s, 1H, 1NH, exchangeable with D_2_O), 11.95 (1br, 2H, 2NH, exchangeable with D_2_O); ^13^C NMR (DMSO-*d_6_*, δ ppm): 55.92 (CH_2_), 122.67, 125.47, 127.77, 129.23, 131.07, 132.98 (aromatic-C), 154.82, 155.06, 167.32, 167.89 (4C=O). MS, *m*/*z* (%): 325 [M^+^ + 1] (23.76), 324 [M^+^] (18.38); Analysis for C_11_H_8_N_4_O_6_S (324.27), Calcd.: %C, 40.74; H, 2.49; N, 17.28; S, 9.89. Found: %C, 40.95; H, 2.65; N, 17.42; S, 10.01.

#### 3.8.5. Preparation of *N*-Substituted-2,3-dioxo-1,2,3,4-tetrahydroquinoxaline-6-sulfonamide Derivatives **7a**–**7c**

A mixture of the sulfonohydrazide compound **3** (2.56 g, 10 mmol) and the appropriate acid anhydride derivatives, namely succinic anhydride, maleic anhydride, and/or phthalic anhydride (10 mmol) in acetic acid (15 mL), was heated under reflux for 8 h. The formed precipitate was filtered, dried, and recrystallized from dioxane to obtain the corresponding compounds **7a**–**c,** respectively.

##### *N*-(2,5-dioxopyrrolidin-1-yl)-2,3-dioxo-1,2,3,4-tetrahydroquinoxaline-6-sulfonamide (**7a**)

Yield (73%); mp. 292–294 °C; IR (KBr, cm^−1^): 3363–3320 (3NH), 3120 (CH, aromatic), 2935 (CH-alicyclic), 1710–1681 (4C=O), 1388, 1180 (SO_2_); ^1^H NMR (DMSO-*d_6_*, δ ppm): 2.73 (s, 4H, 2CH_2_), 7.32, 7.74 (2d, 2H, aromatic-H, *J* = 6.21 Hz), 7.93 (s,1H, aromatic-H), 11.78, 11.83 (2s, 3H, 3NH, exchangeable with D_2_O); ^13^C NMR (DMSO-*d_6_*, δ ppm): 30.07 (2CH_2_, pyrrolidine), 120.82, 124.60, 126.32, 130.25, 132.32, 135.42 (aromatic-C), 154.34, 154.88, 170.81(4C=O); MS, *m*/*z* (%): 339 [M^+^ + 1] (38.73), 338 [M^+^] (20.45); Analysis for C_12_H_10_N_4_O_6_S (338.29), Calcd.: %C, 42.61; H, 2.98; N, 16.56; S, 9.48. Found: %C, 42.53; H, 3.15; N, 16.88; S, 9.24.

##### *N*-(2,5-dioxo-2,5-dihydro-1H-pyrrol-1-yl)-2,3-dioxo-1,2,3,4-tetrahydroquinoxaline-6-sulfonamide (**7b**)

Yield (70%); mp. 295–297 °C; IR (KBr, cm^−1^): 3344–3325 (3NH), 3140 (CH, aromatic), 2943(CH-alicyclic), 1720–1681 (4C=O), 1392, 1134 (SO_2_); ^1^H NMR (DMSO-*d_6_*, δ ppm): 7.20 (d, 2H, 2CH, *J =* 9.04 Hz), 7.56–7.67 (m, 3H, aromatic-H), 11.78, 11.83 (2s, 3H, 3NH, exchangeable with D_2_O); ^13^C NMR (DMSO-*d_6_*, δ ppm): 116.82, 123.72, 126.32, 130.55, 132.48, 135.42, 136.71, 138.42 (aromatic-C), 154.34, 154.88, 165.74 (4C=O); MS, *m*/*z* (%): 322 [M^+^ + 1] (23.68), 320 [M^+^] (20.59); Analysis for C_12_H_8_N_4_O_6_S (336.28), Calcd.: %C, 42.86; H, 2.40; N, 16.66; S, 9.53. Found: %C, 42.53; H, 2.53; N, 16.93; S, 9.70.

##### *N*-(1,3-dioxoisoindolin-2-yl)-2,3-dioxo-1,2,3,4-tetrahydroquinoxaline-6-sulfonamide (**7c**)

Yield (68%); mp. 207–209 °C; IR (KBr, cm^−1^): 3360–3325 (3NH), 3059 (CH, aromatic), 2924 (CH-alicyclic), 1725–1697 (4C=O), 1392, 1138 (SO_2_); ^1^H NMR (DMSO-*d_6_*, δ ppm): 6.88 (d, 1H, aromatic-H, *J* = 6.12 Hz), 7.05−7.12 (m, 3H, aromatic-H), 7.45–7.55(m, 3H, aromatic-H), 9.61, 11.95, 12.01 (3s, 3H, 3NH, exchangeable with D_2_O); ^13^C NMR (DMSO-*d_6_*, δ ppm): 115.13, 123.53, 124.60, 125.28, 126.65, 131.62, 132.00, 132.32, 132.82, 132.93 (aromatic-C), 154.34, 154.88, 170.81 (4C=O); MS, *m*/*z* (%): 386 [M^+^] (28.07); Analysis for C_16_H_10_N_4_O_6_S (386.34), Calcd.: %C, 49.74; H, 2.61; N, 14.50; S, 8.30; Found: %C, 49.97; H, 2.83; N, 14.74; S, 8.48.

#### 3.8.6. Preparation of 2-((2,3-Dioxo-1,2,3,4-tetrahydroquinoxalin-6-yl)sulfonyl)-N-substituted Hydrazine-1-carbothioamide **8a**,**b**

A solution of the sulfonohydrazide derivative **3** (2.56 g, 10 mmol) and the appropriate isothiocyanate, namely 4-methoxybenzene isothiocyanate and/or benzoyl isothiocyanate (10 mmol) in DMF (20 mL) containing a few drops of triethylamine, was heated under reflux for 6 h. After reaction completion, the mixture was poured onto an ice/H_2_O mixture and neutralized with HCl. The formed ppt was collected by filtration, washed several times with water, and recrystallized from ethanol.

##### 2-((2,3-Dioxo-1,2,3,4-tetrahydroquinoxalin-6-yl)sulfonyl)-N-(4-methoxyphenyl) Hydrazine-1-carbothioamide (**8a**)

Yield (73%); mp203–205 °C; IR (KBr, cm^−1^): 3433–3313 (4NH), 3035 (CH, aromatic), 2954 (CH-alicyclic), 1681 (2C=O), 1469 (C=S), 1327, 1172 (SO_2_); ^1^H NMR (DMSO-*d_6_*, δ ppm): 3.81 (s, 3H, OCH_3_), 6.90, 7.31, 7.52, 7.64 (4d, 6H, aromatic-H, *J* = 8.01 Hz), 8.13 (s, 1H, aromatic-H), 10.71, 11.32, 11.97 (3s, 5H, 5NH, exchangeable with D_2_O); ^13^C NMR (DMSO-*d_6_*, δ ppm): 55.44 (CH_3_), 113.63, 113.78, 113.91, 114.01, 114.11, 114.22, 114.27, 118.11, 118.21, 127.09, 134.68, 153.69 (aromatic-C), 155.60 (C=O), 174.98 (C=S); MS, *m*/*z* (%): 422 [M^+^ + 1] (25.38), 421 [M^+^] (17.39); Analysis for C_16_H_15_N_5_O_5_S_2_ (421.45), Calcd.: %C, 45.60; H, 3.59; N, 16.62; S, 15.21; Found: %C, 45.74; H, 3.65; N, 16.85; S, 15.49.

##### *N*-(2-((2,3-dioxo-1,2,3,4-tetrahydroquinoxalin-6-yl)sulfonyl)hydrazine-1-carbonothioyl)benzamide (**8b**)

Yield (75%); mp. 265–267 °C; IR (KBr, cm^−1^): 3460–3417 (4NH), 3093 (CH, aromatic), 2955 (CH-alicyclic), 1795–1681 (3C=O), 1489 (C=S), 1384, 1172 (SO_2_). ^1^H NMR (DMSO-*d_6_*, δ ppm): 7.51–7.75 (m, 3H, aromatic-H), 7.80−8.05 (m, 4H, aromatic-H), 8.14 (s, 1H, aromatic-H), 10.71, 11.32, 11.97 (3s, 5H, 5NH, exchangeable with D_2_O); ^13^C NMR (DMSO-*d_6_*, δ ppm): 115.32, 120.27, 122.73, 125.36, 126.06, 127.39, 129.52, 131.72, 132.35, 133.76, 134.68, 136.27 (aromatic-C), 155.60, 156.58 (2C=O), 180.36 (C=S); MS, *m*/*z* (%): 419 [M^+^] (23.18); Analysis for C_16_H_13_N_5_O_5_S_2_ (419.43), Calcd.: %C, 45.82; H, 3.12; N, 16.70; S, 15.29; Found: %C, 45.97; H, 3.38; N, 16.48; S, 15.36.

#### 3.8.7. Preparation of 4-oxo-2-thioxo-3,4-dihydropyrimidin-1(2H)-yl)-2,3-dioxo-1,2,3,4-tetrahydroquinoxaline-6-sulfonamide Derivatives **9a**, **9b**

A mixture of the thiosemicarbazide derivatives **8a**,**b** (1 mmol) and diethyl malonate (1.6 mL, 1 mmol) in absolute ethanol (20 )mL was refluxed for 10 h. The reaction mixture was cooled and the formed precipitate was filtered, dried, and recrystallized with ethanol to give the target compounds **9a**,**b**, respectively.

##### *N*-(6-hydroxy-3-(4-methoxyphenyl)-4-oxo-2-thioxo-3,4-dihydropyrimidin-1(2H)-yl)-2,3-dioxo-1,2,3,4-tetrahydroquinoxaline-6-sulfonamide (**9a**)

Yield (68%); mp. 205–207 °C; IR (KBr, cm^−1^): 3451 (OH), 3388–3160 (3NH), 3028 (CH, aromatic), 2950 (CH-alicyclic), 1710–1650 (3C=O), 1415 (C=S), 1338, 1196 (SO_2_); ^1^H NMR (DMSO-*d_6_*, δ ppm): 3.72 (s, 3H, OCH_3_), 6.88, 7.10 (2d, 4H, aromatic-H, *J =* 10.01 Hz), 7.22–7.30 (m, 3H, aromatic-H+ pyrimidine-H_5_), 7.45 (d, 1H, aromatic-H, *J =* 10.01 Hz), 9.57 (s, 1H, NH, exchangeable with D_2_O), 10.51 (s, 1H, OH, exchangeable with D_2_O), 11.97 (s, 2H, 2NH, exchangeable with D_2_O); ^13^C NMR (DMSO-*d_6_*, δ ppm): 55.80 (OCH_3_), 82.50 (pyrimidine-C_5_), 115.20, 116.35, 124.28, 125.47, 126.99, 127.30, 127.92, 128.99, 129.34, 131.27, 132.33, 133.48, 135.39 (aromatic–C), 155.47, 155.50 (2C=O), 189.68 (C=S); MS, *m*/*z* (%): 490 [M^+^ + 1] (18.56), 489 [M^+^] (11.49); Analysis for C_19_H_15_N_5_O_7_S_2_ (489.48), Calcd.: %C, 46.62; H, 3.09; N, 14.31; S, 13.10; Found: %C, 46.83; H, 3.15; N, 14.52; S, 13.31.

##### *N*-(3-benzoyl-6-hydroxy-4-oxo-2-thioxo-3,4-dihydropyrimidin-1(2H)-yl)-2,3-dioxo-1,2,3,4-tetrahydroquinoxaline-6-sulfonamide (**9b**)

Yield (70%); mp. 270–272 °C; IR (KBr, cm^−1^): 3449 (OH), 3441–3174 (3NH), 3028 (CH, aromatic), 2924 (CH-alicyclic), 1700–1647 (4C=O), 1415 (C=S), 1338, 1196 (SO_2_); ^1^H NMR (DMSO-*d_6_*, δ ppm): 7.12–7.35 (s, 4H, aromatic-H + pyrimidine-H_5_), 7.62–7.83 (m, 4H, aromatic-H), 8.14 (s, 1H, aromatic-H), 10.51 (s, 1H, OH, exchangeable with D_2_O), 11.98, 12.03 (2s, 3H, 3NH, exchangeable with D_2_O); ^13^C NMR (DMSO-*d_6_*, δ ppm): 82.31 (pyrimidine-C_5_), 116.15, 116.47, 124.61, 126.47, 126.99, 127.92, 128.80, 128.99, 129.12, 130.05, 132.33, 133.05, 133.31 (aromatic–C), 155.44, 155.47, 155.50 (3C=O), 170.1 (C=S); MS, *m*/*z* (%): 488 [M^+^ + 1] (20.84), 487 [M^+^] (16.38); Analysis for C_19_H_13_N_5_O_7_S_2_ (487.46), Calcd.: %C, 46.82; H, 2.69; N, 14.37; S, 13.15. Found: %C, 46.93; H, 2.81; N, 14.52; S, 13.36.

#### 3.8.8. Preparation of *N*′-(3-aryl-4-methylthiazol-2(3H)-ylidene)-2,3-dioxo-1,2,3,4-tetrahydroquinoxaline-6-sulfonohydrazide (**10a**,**b**) and N′-(5-acetyl-3-aryl-4-methylthiazol-2(3H)-ylidene)-2,3-dioxo-1,2,3,4-tetrahydroquinoxaline-6-sulfonohydrazide (**11a**,**b**)

A mixture of compounds **8a**,**b** (1 mmol) and the appropriate α-halo carbonyl compounds (1mmol), namely chloroacetone (0.82 mL) and/or 3-chloroacetylacetone (1.13 mL) in absolute ethanol (30 mL) containing sodium acetate (1.64 g, 20 mmol) was refluxed for 10–12 h. After cooling, the formed precipitate was filtered, washed with water, dried, and crystallized from ethanol to give the target derivatives **10a**,**b**, and **11a**,**b**, respectively.

##### *N*′-(3-(4-methoxyphenyl)-4-methylthiazol-2(3H)-ylidene)-2,3-dioxo-1,2,3,4-tetrahydroquinoxaline-6-sulfonohydrazide (**10a**)

Yield (78%); mp.130–132 °C; IR (KBr, cm^−1^): 3430–3155 (3NH), 3078 (CH, aromatic), 2936 (CH-alicyclic), 1710 (2C=O), 1454 (C=S), 1332, 1180 (SO_2_); ^1^H NMR (DMSO-*d_6_*, δ ppm): 1.85 (s, 3H, CH_3_), 3.80 (s, 3H, OCH_3_), 5.45 (s, 1H, thiazole-H_5_), 6.88, 7.03 (2d, 4H, aromatic-H, *J* = 10.21 Hz), 7.52–7.58 (m, 2H, aromatic-H), 7.84 (s, 1H, aromatic-H), 10.75, 11.96, 12.01 (3s, 3H, 3NH, exchangeable with D_2_O); ^13^C NMR (DMSO-*d_6_*, δ ppm): 18.94 (CH_3_), 55.80 (OCH3), 115.32, 117.26, 118.28, 120.42, 122.41, 125.37, 126.85, 129.06, 129.47, 130.15, 131.75, 132.39, 133.52, 135.29 (aromatic–C), 155.46, 156.74 (2C=O); MS, *m*/*z* (%): 459 [M^+^] (19.30); Analysis for C_19_H_17_N_5_O_5_S_2_ (459.50), Calcd.: %C, 49.67; H, 3.73; N, 15.24; S, 13.95; Found: %C, 49.75; H, 3.84; N, 15.38; S, 14.08.

##### *N*’-(3-benzoyl-4-methylthiazol-2(3H)-ylidene)-2,3-dioxo-1,2,3,4-tetrahydroquinoxaline-6-sulfonohydrazide (**10b**)

Yield (70%); mp. 295–297 °C; IR (KBr, cm^−1^): 3441–3155 (3NH), 3059 (CH, aromatic), 2936 (CH-alicyclic), 1700–1674 (3C=O), 1454 (C=S), 1332, 1180 (SO_2_); ^1^H NMR (DMSO-*d_6_*, δ ppm): 2.33 (s, 3H, CH_3_), 5.43 (s, 1H, thiazole-H_5_), 7.56–7.60 (m, 4H, aromatic-H), 7.66–7.67 (m, 2H, aromatic-H), 8.12 (d, 2H, aromatic-H, *J* = 6.00 Hz), 11.96, 12.01, 12.75 (3s, 3H, 3NH, exchangeable with D_2_O); ^13^C NMR (DMSO-*d_6_*, δ ppm): 19.77 (CH_3_), 116.11, 116.52, 124.60, 126.37, 126.92, 128.76, 129.16, 129.56, 130.15, 132.07, 133.37 (aromatic–C), 155.46, 156.74, 165.56 (3C=O); MS, *m*/*z* (%): 459 [M^+^ + 2] (21.68), 458 [M^+^ + 1] (19.94), 457 [M^+^] (15.49); Analysis for C_19_H_15_N_5_O_5_S_2_ (457.48), Calcd.: %C, 49.88; H, 3.31; N, 15.31; S, 14.02; Found: %C, 49.93; H, 2.94; N, 15.56; S, 13.89.

##### *N*’-(5-acetyl-3-(4-methoxyphenyl)-4-methylthiazol-2(3H)-ylidene)-2,3-dioxo-1,2,3,4-tetrahydroquinoxaline-6-sulfonohydrazide (**11a**)

Yield (75%); mp.185–187 °C; IR (KBr, cm^−1^): 3441–3143 (3NH), 3070 (CH, aromatic), 2946 (CH-alicyclic), 1710–1680 (3C=O), 1469 (C=S), 1332, 1138 (SO_2_); ^1^H NMR (DMSO-*d_6_*, δ ppm): 1.76 (s, 3H, CH_3_), 2.33 (s, 3H, COCH_3_), 3.85 (s, 3H, -OCH_3_), 6.88, 7.10 (2d, 4H, aromatic-H, *J =* 5.67 Hz), 7.24–7.30 (m, 2H, aromatic-H), 7.45 (d, 1H, aromatic-H, *J =* 6.00 Hz), 9.58, 11.96, 12.01 (3s, 3H, 3NH, exchangeable with D_2_O); ^13^C NMR (DMSO-*d_6_*, δ ppm): 14.56 (CH_3_), 25.13 (COCH_3_), 55.70 (OCH_3_), 114.66, 114.95, 115.16, 119.36, 122.05, 126.37, 126.92, 129.16, 129.56, 130.15, 132.07, 133.37 (aromatic-C), 155.68, 176.33 (3C=O); MS, *m*/*z* (%): 502 [M^+^ + 1] (23.44), 501 [M^+^] (14.85); Analysis for C_21_H_19_N_5_O_6_S_2_ (501.53), Calcd.: %C, 50.29; H, 3.82; N, 13.96; S, 12.78; Found: %C, 50.42; H, 3.67; N, 14.08; S, 13.05.

##### *N*’-(5-acetyl-3-benzoyl-4-methylthiazol-2(3H)-ylidene)-2,3-dioxo-1,2,3,4-tetrahydro Quinoxaline-6-sulfonohydrazide (**11b**)

Yield (70%); mp. 290–292 °C; IR (KBr, cm^−1^): 3441–3143 (3NH), 3059 (CH, aromatic), 2924 (CH-alicyclic), 1730–1681 (4C=O), 1469 (C=S), 1332, 1138 (SO_2_); ^1^H NMR (DMSO-*d_6_*, δ ppm): 2.06 (s, 3H, CH_3_), 2.19 (s, 3H, COCH_3_), 7.56–7.60 (m, 3H, aromatic-H), 7.66–7.69 (m, 3H, aromatic-H), 8.12 (d, 2H, aromatic-H, *J =* 10.01 Hz), 11.96, 12.75, (2s, 3H, 3NH, exchangeable with D_2_O); ^13^C NMR (DMSO-*d_6_*, δ ppm): 14.56 (CH_3_), 25.62 (COCH_3_), 116.21, 116.82, 117.37, 118.20, 120.09, 121.83, 125.37, 126.92, 128.52, 129.56, 130.73, 131.85, 133.62, 135.38 (aromatic-C), 155.68, 165.34, 180.27 (4C=O); MS, *m*/*z* (%): 500 [M^+^ + 1] (22.65), 499 [M^+^] (12.37); Analysis for C_21_H_17_N_5_O_6_S_2_ (499.52), Calcd.: %C, 50.50; H, 3.43; N, 14.02; S, 12.84; Found: %C, 50.74; H, 3.78; N, 14.43; S, 12.73.

#### 3.8.9. Preparation of *N*’-(3-Aryl-4-oxothiazolidin-2-ylidene)-2,3-dioxo-1,2,3,4-tetrahydroquinoxaline-6-sulfonohydrazide derivatives 12**a**,**b**

A mixture of compounds **8a**,**b** (1 mmol) and ethyl bromoacetate (1.11 mL, 1 mmol) in absolute ethanol (30 mL) containing sodium acetate (1.64 g, 20 mmol) was refluxed for 8–10 h. After cooling, the formed precipitate was filtered, washed with water, dried, and crystallized from isopropanol to afford the target derivatives **12a**,**b**, respectively.

##### *N*’-(3-(4-methoxyphenyl)-4-oxothiazolidin-2-ylidene)-2,3-dioxo-1,2,3,4-tetrahydro quinoxaline-6-sulfonohydrazide (**12a**)

Yield (69%); mp. 292–294 °C; IR (KBr, cm^−1^): 3441–3417 (3NH), 3059 (CH, aromatic), 2924 (CH-alicyclic), 1710, 1650 (3C=O), 1415 (C=S), 1332, 1138 (SO_2_); ^1^H NMR (DMSO-*d_6_*, δ ppm): 3.72 (s, 2H, thiazolidine-CH_2_), 4.15 (s, 3H, OCH_3_), 7.11, 7.21 (2d, 4H, aromatic-H, *J* = 6.53 Hz), 7.23, 7.28 (2d, 2H, aromatic-H, *J* = 5.67 Hz), 8.52 (s, 1H, aromatic-H), 9.54, 11.97, 12.02 32s, 3H, 3NH, exchangeable with D_2_O); ^13^C NMR (DMSO-*d_6_*, δ ppm): 25.13 (thiazolidine-CH_2_), 63.09 (OCH_3_), 114.66, 114.96, 115.16, 119.36, 122.03, 128.80, 129,10, 135.50, 136.38, 155.68 (aromatic-C), 163.27, 176.33 (3C=O); MS, *m*/*z* (%): 462 [M^+^ + 1] (23.53), 461 [M^+^] (15.09); Analysis for C_18_H_15_N_5_O_6_S_2_ (461.47), Calcd.: %C, 46.85, H, 3.28; N, 15.18; S, 13.89. Found: %C, 46.76; H, 2.95; N, 14.74; S, 13.62.

##### *N*’-(3-benzoyl-4-oxothiazolidin-2-ylidene)-2,3-dioxo-1,2,3,4-tetrahydroquinoxaline-6-sulfonohydrazide(**12b**)

Yield (69%); mp. 290–292 °C; IR (KBr, cm^−1^): 3441–3417 (3NH), 3068 (CH, aromatic), 2995 (CH-alicyclic), 1710–1675 (4C=O), 1415 (C=S), 1332, 1138 (SO_2_); ^1^H NMR (DMSO-*d_6_*, δ ppm): 4.32 (s, 2H, thiazolidine-CH_2_), 7.52–7.72 (m, 4H, aromatic-H), 7.85–8.06 (m, 3H, aromatic-H), 8.25 (s, 1H, aromatic-H), 9.79, 10.50 (2s, 3H, 3NH, exchangeable with D_2_O); ^13^C NMR (DMSO-*d_6_*, δ ppm): 30.52 (thiazolidine-CH_2_), 117.28, 120.39, 123.20, 125.65, 129,46, 130.62, 133.42, 135.50, 136.38, 139.57, 143.27 (aromatic-C), 155.78, 163.27, 176.33 (4C=O); MS, *m*/*z* (%): 460 [M^+^ + 1] (24.56), 459 [M^+^] (12.37); Analysis for C_18_H_13_N_5_O_6_S_2_ (459.45), Calcd.: %C, 47.06, H, 2.85; N, 15.24; S, 13.96; Found: %C, 46.85; H, 2.95; N, 15.38; S, 13.71.

## 4. Conclusions

The current study deals with the design and synthesis of a novel set of derivatives **3**–**12a**,**b** bearing the quinoxaline scaffold that is hybridized with various heterocyclic ring systems via a sulfonamide linkage. The new compounds were assessed for their suppression impact against the PARP-1 enzyme using Olaparib as a positive reference drug. Among the examined compounds, **4**, **5**, **8a**, **10b**, and **11b** displayed the highest PARP-1 inhibitory suppression effect with IC_50_ values ranging from 2.31 to 8.25 nM, compared to IC_50Olaparib_ of 4.40 nM. The latter compounds were further examined as antiproliferative agents in MDA-MB-436 in comparison with Olaparib as a reference drug. The compounds **5**, **8a**, **10b**, and **11b** exhibited promising inhibitory activity with IC_50_ values ranging from 2.57 to 11.50 µM, compared to IC_50Olaparib_ of 4.40 µM, and confirmed a safety profile against the normal cells’ WI-38 cell lines. Due to the well-balanced activity of compound **5** as a promising PARP-1 inhibitor, as well as the antiproliferative agent, it was chosen as a representative example for further cellular mechanistic investigation regarding its impact on the cell cycle progression and induction of apoptosis in the MDA-MB-436 cell line. Treatment of the latter cells with compound **5** led to cell cycle arrest at the G2/M phase and demonstrated apoptotic and necrotic effects in comparison to the untreated control cells and increased the autophagic cell death (68.65%).

Molecular docking of the newly synthesized hybrids **4**, **5**, **8a**, **10b**, and **11b** in the PARP-1 active sites involved their good accommodation interacting with the various amino acid residues through hydrogen bonding and π-π contacts. The SwissADME tool represented the good GIT absorption of compound **5** and the inability of all the compounds to cross the BBB.

## Data Availability

Not applicable.

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
