# Peer review of "New Quinoxaline-Based Derivatives as PARP-1 Inhibitors: Design, Synthesis, Antiproliferative, and Computational Studies"

_molecules, 2022, doi:10.3390/molecules27154924_

Round 1

Reviewer 1 Report

In this current submission of the manuscript, the author described the synthesis, bioactivity and computational studies of quinoxaline-based derivatives as PARP-1 inhibitors. They have synthesized some quinoxaline-based sulfonyl hydrazide derivatives using some known literature methods and further studied biological evaluation (IC50 values, cell cycle analysis, Apoptosis and Autophagy assay).  The results are quite interesting and suitable for publication in Molecules. However, some minor revisions are needed before final acceptance of the manuscript.

Comments:

1.      please write down yield of the product in every step of the reaction in Scheme 1 and 2 for better understanding to the reader (above or below of the arrow)

2.      It is difficult for readers to understand product structure with name provided (7a-c, 10a,b, etc…). Please mentioned proper notation like 7a and 7b and so on separately with their structure in every figure and scheme (e.g. 8a; Ar = 4-OMe-C6H4).

3.      Scheme 2, ph-CO is wrong. It should be PhC(O)

4.      Author mentioned sulfonamide group in para position of quinoxaline provided good biological activities. However, all other PARP-1 inhibitors presented in figure 1 are Carboxamide (CONH) based drug. Did author check this study with Carboxamide base quinoxaline derivative?

5.      In manuscript, author provided all the spectral data (1H and 13C NMR) of synthesized compounds, however some spectra in the supporting information are missing. Please include those spectra in supporting information (13C NMR of compound 4,9a, 11b, 12a and 1H NMR of 8a, 9b).

Author Response

Reviewer #1:

  1. please write down the yield of the product in every step of the reaction in Schemes 1 and 2 for better understanding to the reader (above or below the arrow).

The yields of the products in every step of the reaction in Schemes 1 and 2 were added below the arrows.

  1. It is difficult for readers to understand product structure with the name provided (7a-c, 10a,b, etc…). Please mentioned proper notations like 7a and 7b and so on separately with their structure in every figure and scheme (e.g. 8a; Ar = 4-OMe-C6H4).

The required correction was performed (please see the figure and scheme attached).

  1. Scheme 2, ph-CO is wrong. It should be PhC(O).

ph-CO was corrected to PhC(O) in Scheme 2.

  1. The author mentioned sulfonamide group in para position of quinoxaline provided good biological activities. However, all other PARP-1 inhibitors presented in figure 1 are Carboxamide (CONH) based drugs. Did the author check this study with Carboxamide base quinoxaline derivative?

It has been documented that PARP-1 suppressors shared common pharmacophoric features which are an aromatic ring and a carboxamide core. Also, plenty of research represented that restriction of the carboxamide-free rotation greatly enhances the PARP1 inhibitory activity. The quinoxaline nucleus possesses the basic scaffold of the plurality of PARP-1 inhibitors bearing two restricted carboxamides.

All the newly synthesized compounds of this work bear the basic pharmacophoric features of PARP-1 inhibitors which are an aromatic ring and a carboxamide core which is the 1,4- quinoxaline nucleus in this study. All the compounds were attached to different side chains via SO2 which was selected as a linker due to its reported anticancer activity. This means that all of the compounds have the basic requirements to inhibit PARP-1 enzyme not the SO2 group at the 6-position of the quinoxaline ring. Accordingly, the variation in PARP-1 inhibitory activity in this study depends on the type and the length of the side chain attached to the sulfonyl group. The results showed the attachment of 3,5-dimethylpyrazole ring and 4-methoxyphenyl hydrazinecarbothioamide (compounds 5 and 8a, respectively) to the basic scaffold 6-sulfonyl-1,4- quinoxaline exhibited the most potent PARP-1 inhibition. Consequently, the attachment of 3-benzoyl-4-methylthiazoline and 5-acetyl- 3-benzoyl-4-methylthiazoline moieties via the hydrazide linker to the basic 6-sulfonoquinoxaline (congeners 10b and 11b) produced a detectable reduction in the activity. 

All the designed compounds have promising PARP-1 inhibitory activity since all of them bear the basic features of the target activity. On the other hand, the degree of the potency depends on the type and the length of the side chain attached to the quinoxaline ring via the SO2 group.

  1.  In the manuscript, the author provided all the spectral data (1H and 13C NMR) of synthesized compounds, however, some spectra in the supporting information are missing. Please include those spectra in supporting information (13C NMR of compounds 4,9a, 11b, 12a and 1H NMR of 8a, 9b).

Figures were added to the supporting information

Reviewer 2 Report

The authors have done a wonderful job of presenting the use of the Quinoxaline scaffold in designing new PARP-1 inhibitors. By comparing the activity with Olaparib, the authors found 5 new chemical entries with very promising PARP-1 inhibition activity.

This is a very nice piece of work, a very joyful reading experience, and will likely be well cited by the community.

Additional questions:

1.     compounds 8a, 5 were the most promising suppressors compared to Olaparib. Compounds 4, 10b, and 11b showed a mild decrease in the potency of the IC50 range.

Why not test the real solubility of the compounds?

2.     Why not test the compounds pKa, which is the key impediment to oral availability?

3.     Did the authors ever consider taking cellular sensitization essay, PF50?

Author Response

Reviewer #2:

This is a very nice piece of work, a very joyful reading experience, and will likely be well cited by the community.

Additional questions:

  1. compounds 8a, and 5 were the most promising suppressors compared to Olaparib. Compounds 4, 10b, and 11b showed a mild decrease in the potency of the IC50 range.

Why not test the real solubility of the compounds?

  1. Why not test the compounds pKa, which is the key impediment to oral availability?
  2. Did the authors ever consider taking cellular sensitization essay, PF50?

We thank the reviewer for his positive feedback and encouragement. As part of our lead optimization strategy, we are planning to assess the in vitro and in vivo pharmacokinetic properties. Further optimized structures will be synthesized, and the cellular sensitization assay will be considered.
